# The Intake of Kiwifruits Improve the Potential Antioxidant Capacity in Male Middle- and Long-Distance Runners Routinely Exposed to Oxidative Stress in Japan

**DOI:** 10.3390/sports9030037

**Published:** 2021-03-03

**Authors:** Nami Imai, Yuki Kobayashi, Kazuhiro Uenishi

**Affiliations:** Laboratory of Physiological Nutrition, Kagawa Nutrition University, 3-9-21 Chiyoda, Sakado-shi, Saitama 350-0288, Japan; imai.nami@eiyo.ac.jp (N.I.); kya181@eiyo.ac.jp (Y.K.)

**Keywords:** athletes, fruit, antioxidant activity, d-ROMs, BAP

## Abstract

Oxidation damages cells and muscles, and thus, causes injuries and fatigue, which negatively affect the conditioning of athletes. Thus, in this study, we aimed to investigate the effects of high-antioxidant fruits (kiwifruit) intake on oxidative stress level (d-ROMs) and antioxidant activity (BAP) in male middle- and long-distance runners routinely exposed to oxidative stress. This study was performed from May to July 2017 (Study 1) and October to December 2018 (Study 2). The subjects in Study 1 were 30 male runners, of which 15 consumed two yellow kiwifruits (Zespri^®^ SunGold Kiwifruit) per day for one month of the survey period (Intake group). The subjects of Study 2 were 20 male runners who had high d-ROMs from preliminary testing. These runners consumed two yellow kiwifruits (Zespri^®^ SunGold Kiwifruit) per day for two months. d-ROMs and BAP were measured using a free radical analyzer. In study 1, the d-ROMs decreased while the potential antioxidant capacity (BAP/d-ROMs ratio) increased in the Intake group. In study 2, BAP/d-ROMs ratio was higher after one and two months compared to that at pre-intervention. Study findings suggested that consumption of kiwifruits may reduce oxidative stress levels and increase antioxidant activity, resulting in improved potential antioxidant capacity.

## 1. Introduction

Oxygen consumption in the body during exercise is considered to reach about 100 times that at rest [1]. Therefore, athletes who routinely train at high intensity take in a large amount of oxygen into their bodies every day. An increase of oxygen intake also increases the generation of free radicals in the body, leading to an oxidative state. Oxidation damages cells and muscles, and thus, causes injuries and fatigue, which negatively affect the conditioning of athletes. Therefore, preventing oxidation is an important prerequisite for successful conditioning.

Reactive oxygen species (ROS) are highly reactive oxygen compounds [2]. These include superoxide (O_2_·^−^), hydrogen peroxide (H_2_O_2_), hydroxyl radical (HO·) and singlet oxygen (^1^O_2_). Since active oxygen reacts promptly, it causes cytotoxicity due to inactivation of oxygen, lipid peroxidation and damage to nucleic acids. Our body is provided with an antioxidant defense system that remove ROS [3]. The state in which the production of ROS and free radicals exceeds antioxidant activity is called “oxidative stress”. Furthermore, oxidative stress attributed to exercise is caused by an increase in active oxygen associated with an increase in oxygen intake. ROS production during exercise is a time-dependent increase and increase dependent on exercise intensity [4]. To inhibit the generation of active oxygen and free radicals, it is believed that endogenous antioxidant enzymes and intake of antioxidant substances, such as vitamin C, vitamin E and β-carotene, are essential [5]. Vitamin E is a fat-soluble vitamin that is present in biological membranes and removes ROS produced in biological membranes. Vitamin C is a water-soluble vitamin that is present in the cytoplasm and extracellularly and removes ROS produced outside the biological membrane. It has been reported that plasma vitamin C levels reflect dietary intake [6] and vitamin C inhibits lipid peroxidation [7]. Therefore, the intake of antioxidant substances, such as vitamins C and E, reduces oxidative stress [8]. However, it has also been reported that such intake from high-dose supplements negatively affects exercise performance [9,10,11,12,13]. Complexities continue to surround the relationship between exercise and the intake of antioxidant supplements. Further reports have suggested that the intake of antioxidative substances from food products has no negative effect [14]; hence, it is considered imperative for antioxidative substances to be consumed in food products rather than in supplements.

Fruits are one of the foods that are high in antioxidants. Among the fruits, kiwifruit has a high content of vitamin C and vitamin E (Table 1). In addition, yellow kiwifruit has a higher content of vitamin C and vitamin E than green kiwifruit and Zespri^®^ SunGold Kiwifruit has the highest content of vitamin C (Table 1). It was reported that ingestion of kiwifruit increased plasma vitamin C levels and increased antioxidant capacity [15,16]. We considered appropriate to use kiwifruit as a test food. Further, we decided to use Zespri^®^ SunGold Kiwifruit, which has a high vitamin C content, as a test food. Another reason is that it is relatively easy to eat and can be eaten raw.

Thus, in this study, we aimed to investigate the effects of kiwifruit intake on oxidative stress level and antioxidant activity in male middle- and long-distance runners routinely exposed to oxidative stress.

## 2. Materials and Methods

### 2.1. Subjects

This study was performed on the A University track and field team from May to July 2017 (Study 1) and October to December 2018 (Study 2). The subjects were elite level among universities. Therefore, they continued to run during the investigation. The subjects in Study 1 were 30 male middle- and long-distance runners (20.5 ± 0.8 years old), of which 15 consumed two yellow kiwifruits (Zespri^®^ SunGold Kiwifruit) per day for one month of the survey period (Intake group). The remaining 15 runners from Study 1 had a regular diet with no additional kiwifruit (Control group). The Zespri^®^ SunGold Kiwifruit is high in antioxidants such as Vitamin C and it was reported that the intake of two yellow kiwifruits per day by individuals with moderate mood disturbance can improve overall mood [19]. Thus, we also decided to eat two yellow kiwifruits (Zespri^®^ SunGold Kiwifruit; weight: 162 g. Energy 96 kcal, Vitamin C 261 mg, Vitamin E 2.3 mg) per day in the take group. We did not strictly control their diet because of the negative burden involved in doing so. However, they lived in the same dormitory and breakfast and dinner were unified menus. Therefore, it was considered that the subjects’ diets were similar. In addition, it was considered that the subjects did not lack nutrients intake without the kiwifruit. Because the subjects’ breakfast and dinner were controlled by a registered dietitian. Few runners ingested any dietary supplements fortified with antioxidant nutrients.

We performed an additional investigation to examine the effect for runners who had high oxidative stress levels (Study 2). The subjects of Study 2 were 20 male long-distance runners (20.4 ± 1.0 years old) who had high d-ROMs values from preliminary testing performed in September 2018. These runners consumed two yellow kiwifruits (Zespri^®^ SunGold Kiwifruit) per day for two months.

This study was approved by the Research Ethics Review Committee of Kagawa Nutrition University (No. 114 and 181). All subjects gave their informed consent for inclusion before they participated in the study.

### 2.2. Anthropometry

Height was measured using a height meter with a precision of 0.1 cm. Weight, lean body mass and body fat percentage were measured using InBody720 (Inbody Japan Co., Tokyo, Japan). Body mass index (BMI) was calculated using the following Equation:BMI (kg/m^2^) = Weight (kg)/[Height (m)]^2^(1)

### 2.3. Measurement of Oxidative Stress Markers

Diacron-reactive oxygen metabolites (d-ROMs) was used as an index of oxidative stress [20] and biological antioxidant potential (BAP) was used as an index of antioxidant activity [21]. The d-ROMs assess hydroperoxides that are the products of dehydrogenation and peroxidation of several cellular components, including lipids, proteins, amino acids and nucleic acids. The BAP assesses the reducing power of endogenous antioxidant substances (e.g., proteins, bilirubin and uric acids) and extrinsic antioxidant substances (e.g., vitamin C, vitamin E and polyphenol) against reactive species.

Blood was collected on an empty stomach in the morning. In Study 1, it was conducted twice before and after the intervention and in Study 2, it was conducted three times before, one month and two months after the intervention. After blood collection, the blood was immediately centrifuged (3000 rpm, 10 min) and the serum was cryopreserved. Frozen serum was thawed when d-ROMs and BAP measurements were taken (Study 1, July 2017; Study 2, December 2018). d-ROMs and BAP were measured using a free radical analyzer (FREE Carrio Duo, Wismerll Co., Tokyo, Japan).

To measure d-ROMs, we placed 20 µL of serum in a cuvette containing a pH 4.8 acidic buffer and mixed by inversion. When a serum sample is dissolved in an acidic buffer, the hydroperoxides react with the free irons (Fe_2_^+^ and Fe_3_^+^) and are converted to alkoxyl (R-O·) and peroxyl (R-OO·) radicals. Further, we placed 20 µL d-ROMs coloring solution (colorless aromatic amine solution, N,N-diethyl-para-phenylenediamine) in a cuvette and mixed by inversion. Alkoxyl (R-O·) and peroxyl (R-OO·) radicals oxidized N,N-diethyl-para-phenylenediamine (A-NH_2_) and converted it to a colored derivative. It was set in a free radical analyzer and the amount of change in absorbance at a wavelength of 505 nm was calculated after 5 min. According to the manufacturer’s, simultaneous reproducibility is 2.07 (CV %) and day difference reproducibility is 1.79 (CV %). d-ROMs evaluates the hydroperoxide (ROOH) concentration and 1 U.CARR has an oxidizing ability equivalent to 0.08 mgH_2_O_2_/dL. A high d-ROMs value indicated having high oxidative stress and values > 300 U.CARR indicate a condition of oxidative stress [22] (the manufacturer’s reference values were as follows: 200 ≤ Normal ≤ 300, 300 < Borderline ≤ 320, 320 < Mild oxidative stress ≤ 340, 340 < Medium oxidative stress ≤ 400, 400 < Strong oxidative stress ≤ 500, 500 < Very Strong oxidative stress).

To measure BAP, we placed 50 µL BAP coloring solution in a cuvette and mixed by inversion. It was set in a photometer and measured the absorbance at 505 nm. Then, we placed 10 µL serum in a cuvette, mixed by inversion, set in a photometer, measured the absorbance and calculated the change in absorbance. FeCl_3_ contained in the coloring solution turns red as Fe^3+^ when dissolved in a colorless solution containing a thiocyanate derivative (AT). When a serum sample is added there, it is reduced to Fe^2+^ by the action of antioxidants in the sample and decolorized. This color change is measured by a photometer and BAP evaluates the oxidized iron ions concentration. According to the manufacturer’s, simultaneous reproducibility is 2.15 (CV %) and day difference reproducibility is 3.05 (CV %). A high BAP value indicated a high defense against oxidative stress and values are ≥ 2200 mol/L for an optimum status [21,23] (the manufacturer’s reference values were as follows: Optimal ≥ 2200, 2200 > Borderline ≥ 2000, 2000 > Slight lack of antioxidant activity ≥ 1800, 1800 > Lack of antioxidant activity ≥ 1600, 1600 > Considerable lack of antioxidant activity ≥ 1400, 1400 > Significant lack of antioxidant activity).

BAP/d-ROMs ratio used as an index of potential antioxidant capacity [23]. The criterion value for the BAP/d-ROMs ratio was set at 7.3. Accordingly, a value lower than 7.3 was defined as an oxidized type and a higher one as a reduced type. A higher BAP/d-ROMs was considered preferable.

### 2.4. Serum Creatine Kinase (CK) and Lactate Dehydrogenase (LDH)

Serum CK and LDH were used as indexes of muscle damage. Blood was collected on a similar schedule and measured each time before cryopreservation. The measurement was outsourced to the company (SRL, Inc., Tokyo, Japan).

### 2.5. Statistical Analysis

All data were statistically analyzed using SPSS version 20 (IBM, Japan Inc., Tokyo, Japan). The level of significance was set at *p* < 0.05. The normality of the distributions was verified using the Shapiro–Wilk test. In Study 1, for the relationship between oxidative stress markers and measured values, the spearman’s rank correlation coefficient was used. For purposes of comparing the Intake group and Control group, the unpaired *t*-test was used for normal distribution data and the Mann–Whitney U-test for non-normal distribution data. To compare pre-intervention with post-intervention data, the paired *t*-test was used for normal distribution data and the Wilcoxon signed rank test for non-normal distribution data. In Study 2, for the comparison of data at 3 time points (pre-intervention, after a month, after two months), repeated measure analysis of variance (ANOVA) was used for normal distribution data and Friedman’s test for non-normal distribution data. Bonferroni’s correction was used for multiple comparisons.

## 3. Results

### 3.1. Characteristics of Subjects

The characteristics of subjects are shown in Table 2. The subjects are middle- and long-distance runners and their body fat percentage is low. In both studies, d-ROMs was in the “normal” range and BAP was in the “borderline” range.

### 3.2. Relationship between Oxidative Stress Markers and Measured Values in Study 1

Oxidative stress markers (d-ROMs, BAP and BAP/d-ROMs ratio) were not associated with anthropometric values (height, weight, BMI, body fat percentage and lean body mass) in Study 1 (all *p* > 0.05). In addition, oxidative stress markers (d-ROMs, BAP and BAP/d-ROMs ratio) were not associated with CK and LDH in Study 1 (all *p* > 0.05).

### 3.3. Changes in Study 1

The comparisons between pre- and post-intervention values in the two groups of Study 1 are shown in Table 3. There was no significant difference in d-ROMs, BAP and BAP/d-ROMs ratio between the Control and Intake groups at pre-intervention. The CK and LDH were higher in the Intake group than in the Control group at pre-intervention. However, there was no significant difference between the Control and Intake groups at post-intervention. On comparing pre- and post-intervention values in the Control group, no significant changes emerged in all endpoints. In the Intake group, d-ROMs decreased while the BAP/d-ROMs ratio increased.

### 3.4. Changes in Runners Having High d-ROMs Values (Study 2)

A significant decrease in d-ROMs was observed in the Intake group. Thus, an intervention study, in which the kiwifruits were consumed by the runners having high d-ROMs values, was performed the following year (Study 2). The results of d-ROMs, BAP, BAP/d-ROMs ratio, CK and LDH in Study 2 are shown in Figure 1. The d-ROMs values changed from 293 (275–310) U.CARR to 268 (248–283) U.CARR and 280 (262–291) U.CARR. The BAP values changed from 2127 (2087–2216) µmol/L to 2207 (2069–2241) µmol/L and 2246 (2179–2338) µmol/L. Thus, the BAP/d-ROMs ratio changed from 7.5 (6.9–7.8) to 8.0 (7.5–8.7) and 8.2 (7.4–8.8), which was higher after one and two months compared to that at pre-intervention (Pre). The CK values changed from 277 (201–467) U/L to 757 (465–1044) U/L and 468 (307–620) U/L, which was higher after one month compared to that at pre-intervention (Pre). The LDH values changed from 220 (197–259) U/L to 241 (224–295) U/L and 238 (207–263) U/L, which was higher after one month compared to that at pre-intervention (Pre).

## 4. Discussion

This study aimed at investigating the effects of high-antioxidant fruits (kiwifruit) intake on oxidative stress and antioxidant activity in male middle- and long-distance runners routinely exposed to oxidative stress in Japan. The results of this study suggested that the intake of kiwifruits may improve the potential antioxidant capacity.

Oxidative stress has been investigated from various perspectives. The markers such as 8-iso-prostaglandin-F2α (lipid peroxidation markers) and 8-hydroxydeoxyguanosine (8-OHdG, DNA oxidative damage markers) are often used and are useful [24]. However, these assay methods are complex and slow. The d-ROMs and BAP are measured simpler and faster. d-ROMs assess hydroperoxides that are the products of including lipids peroxidation and BAP assesses the reducing power against reactive species. Thus, in this study, we measured d-ROMS and BAP as markers of oxidative stress and antioxidant capacity.

The d-ROMs values were 264 ± 34 U.CARR, BAP values were 2036 ± 145 µmol/L, BAP/d-ROMs ratio were 7.9 ± 1.2 in Study 1′ all subjects. In Sone’s study, Japanese elite male university athletes had a d-ROMs value of 271 ± 42 U.CARR, BAP value of 2154 ± 143 µmol/L and BAP/d-ROMs ratio of 8.13 ± 1.27 [25]. The BAP/d-ROMs ratio was slightly lower in our study subjects. Oxidative stress markers (d-ROMs, BAP and BAP/d-ROMs ratio) were not associated with anthropometric values (height, weight, BMI, body fat percentage and lean body mass), so oxidative stress was unaffected by body composition. In addition, oxidative stress markers were not associated with muscle damage (CK and LDH). Serum creatine kinase (CK) and lactate dehydrogenase (LDH) were indexes of muscle damage [26]. In addition, it has been reported that d-ROMs caused by exercise was positively associated with leukocyte count and muscle damage markers such as myoglobin [27]. However, oxidative stress markers were independent of muscle damage in this study.

In Study 1, d-ROMs values in male middle- and long-distance runners were 249 (225–286) U.CARR in the Control group and 270 ± 35 U.CARR in the Intake group. These values in both groups were “normal.” The BAP values were 2059 ± 142 µmol/L in the Control group and 2013 ± 149 µmol/L in the Intake group. These values were “borderline.” It has been reported that exercise training increases antioxidant-enzyme levels in limb skeletal muscles and diaphragm [28]. However, it is conceivable that an active oxygen system may overtake an antioxidant defense function due to increased oxygen intake for consecutive days. Therefore, the balance between oxidative stress and antioxidant activity in athletes is considered an important issue. BAP/d-ROMs ratio was 8.1 ± 1.2 in the Control group and 7.1 (6.9–8.2) in the Intake group. The Control group was the reduced type, while the Intake group was the oxidized type. However, no adjustment was made because there was no significant difference between the two groups.

On comparing pre- and post-intervention data from the Control group, there was no significant change. However, in the Intake group, d-ROMs values decreased to 247 ± 25 and BAP/d-ROMs ratio increased to 8.7 (7.3–9.8). BAP/d-ROMs ratio in the Intake group improved to a reduced type. Due to maintenance of BAP values, reduced oxidative stress increased the potential antioxidant capacity. It is reported that the levels of markers of muscle lipid and protein oxidation are reduced by training while antioxidant enzymes activity are increased [29]. In the case of trained rats, it is also reported that acute exercise cannot be expected to further increase antioxidant activity [28]. Thus, the d-ROMs may be reduced by the adaptive muscle responses of regular training. However, in this study, the subjects (the intake group and the control group) were running even pre-intervention and continued to run during the intervention. It is difficult to think that the d-ROMs may have reduced by the adaptive muscle responses of regular training in this Study. The increase in active oxygen is prevented by endogenous antioxidative enzymes and antioxidant substances, such as vitamin C, vitamin E and β-carotene [5]. Kiwifruit contains vitamin C, vitamin E and carotenes in abundance. The additional nutrients (two kiwifruits) in the intake group were energy 96 kcal, vitamin C 261 mg and Vitamin E 2.3 mg per day. It was reported that intake of 500 mg vitamin C inhibits lipid peroxidation [7]. Muscle damage markers of CK and LDH reduces by inhibiting lipid peroxidation [7]. Moreover, it was reported that oxidative activity was not promoted by low dose intake (vitamin A 400 µg, vitamin E 15 mg and vitamin C 30 mg) [22]. In the intake group of this study, they ingested more vitamin C by consuming kiwifruit. As a result, it is considered that lipid peroxidation was inhibited. Because d-ROMs reflect the products of including lipid peroxidation. Thus, it is conceivable that the intake of additional kiwifruits led to increased intake of antioxidant substances, which reduced oxidative stress and improved the balance between oxidative stress and antioxidant activity. In addition, their consumption in food products was of greater benefit than that in high dose supplements.

The possible effect of reducing d-ROMs in the Intake group was suggested in Study 1. Thus, an additional investigation by the runners who had high d-ROMs values was performed (Study 2). During the period of kiwifruit intake, d-ROMs initially decreased and BAP gradually increased. The d-ROMs after two months remained unchanged compared to those at pre-intervention. However, the BAP/d-ROMs ratio remained higher than that at pre-intervention after one month as well as after two months. The CK and LDH after a month were higher than pre-intervention, showing the opposite change to d-ROMs. Since CK and LDH were indexes of muscle damage, it is considered that CK and LDH increased by high exercise loads. Although the exercise loads have not been investigated, it is considered that antioxidant mechanism withstands high exercise loads. Therefore, it is conceivable that antioxidant mechanism may be corrected and BAP/d-ROMs may increase or remain constant by continuous intake of kiwifruits.

The limitation of this study is that we neither considered an individual’s frequency and amount of practice nor investigated their activity and sleep durations. Hence, further study is needed. In this study, we did not perform detailed nutritional assessment of the subjects. Because it was their negative burden. In addition, we did not measure micronutrients of actual kiwifruit ingested and vitamin C and vitamin E in blood. These measurements should be measured in future studies to improve the reproducibility of the study. If we evaluate their dietary intake and supplement intake, measure actual kiwifruit ingested, vitamin C and vitamin E in blood, we can discuss vitamin C and vitamin E more clearly. Moreover, women were reported to have higher d-ROMs than men did [25,30]. Thus, women may produce different results than those of men.

Study findings suggested that consumption of high-antioxidant fruits (2 yellow kiwifruits per day) may reduce oxidative stress (d-ROMs) and increase antioxidant activity (BAP), resulting in improved potential antioxidant capacity (BAP/d-ROMs ratio) in male middle- and long-distance runners routinely exposed to oxidative stress.

## Figures and Tables

**Figure 1 sports-09-00037-f001:**
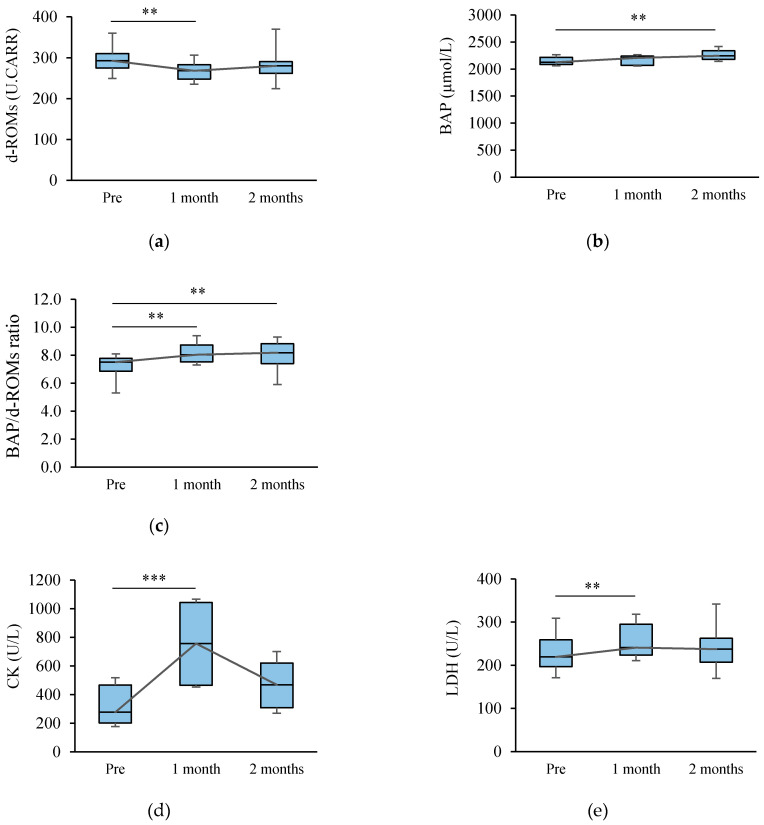
Changes of oxidative stress markers, CK and LDH in Study 2. (**a**) Changes in diacron-reactive oxygen metabolites (d-ROMs) during the survey period (n = 20); (**b**) Changes in biological antioxidant potential (BAP) during the survey period (n = 20); (**c**) Changes in BAP/d-ROMs ratio during the survey period (n = 20); (**d**) Changes in creatine kinase (CK) during the survey period (n = 20); (**e**) Changes in lactate dehydrogenase (LDH) during the survey period (n = 20). Statistics: ** *p* < 0.01, *** *p* < 0.001. d-ROMs, BAP/d-ROMs ratio and CK were used Friedman test (Bonferroni), BAP and LDH were used repeated measure ANOVA (Bonferroni). Box plot: (minimum values)-(25 percentile)-median-(75 percentile)-(maximum values). Pre: pre-intervention, 1 month: after a month, 2 months: after two months.

**Table 1 sports-09-00037-t001:** Nutrient contents of fruits (per 100 g).

	Zespri^® †^		Food Composition Table ^§^				
	Green Kiwifruit	SunGold Kiwifruit	Green Kiwifruit	Yellow Kiwifruit	Strawberry	Blueberry	Orange
Energy (kcal)	53	59	53	59	34	49	39
Protein (g)	1.2	1.0	1.0	1.1	0.9	0.5	1.0
Carbohydrate (g)	14.0	15.8	13.5	14.9	8.5	12.9	9.8
Dietary fiber (g)	3.0	1.4	2.5	1.4	1.4	3.3	0.8
Water-soluble dietary fiber (g)	0.6	0.4	0.7	0.5	0.5	0.5	0.3
Insoluble dietary fiber (g)	2.4	1.1	1.8	0.9	0.9	2.8	0.5
Calcium (mg)	27	17	33	17	17	8	21
Iron (mg)	0.2	0.2	0.3	0.2	0.3	0.2	0.3
Magnesium (mg)	14	12	13	12	13	5	11
Potassium (mg)	301	315	290	300	170	70	140
Zinc (mg)	0.1	0.1	0.1	0.1	0.2	0.1	0.2
Vitamin C (mg)	85	161	69	140	62	9	40
Vitamin B_1_ (mg)	0.00	0.00	0.01	0.02	0.03	0.03	0.10
Vitamin B_2_ (mg)	0.05	0.07	0.02	0.02	0.02	0.03	0.03
Niacin (mg)	0.0	0.2	0.3	0.3	0.4	0.2	0.4
Pantothenic acid (mg)	0.00	0.12	0.29	0.26	0.33	0.12	0.36
Vitamin B_6_ (mg)	0.07	0.08	0.12	0.14	0.04	0.05	0.07
Folic acid (µg)	38	31	36	32	90	12	32
Vitamin B_12_ (µg)	0.0	0.1	0.0	0.0	0.0	0.0	0.0
Vitamin A, Retinol equivalent (µg)	9	2	6	3	1	5	10
Vitamin E, α-Tocopherol (mg)	0.9	1.4	1.3	2.5	0.4	1.7	0.3

^†^: Reprinted with permission from ref. [17]. 2018 Zespri International (Japan) K.K., ^§^: Reprinted with permission from ref. [18]. 2015 Ministry of Education, Culture, Sports, Science and Technology.

**Table 2 sports-09-00037-t002:** Characteristics of subjects.

	Study 1 (n = 30)	Study 2 (n = 20)
Height (cm)	172.7 ± 4.4	171.3 ± 4.1
Weight (kg)	58.0 ± 3.8	57.9 ± 4.0
BMI (kg/m^2^)	19.4 ± 0.9	19.7 ± 0.8
Body fat percentage (%)	10.9 ± 2.7	10.7 ± 2.5
Lean body mass (kg)	51.7 ± 4.1	51.8 ± 3.8
d-ROMs (U.CARR)	264 ± 34	293 ± 25
BAP (µmol/L)	2036 ± 145	2123 ± 100
BAP/d-ROMs ratio	7.9 ± 1.2	7.3 ± 0.7
CK (U/L)	584 ± 415	335 ± 167
LDH (U/L)	239 ± 40	230 ± 46

Data are pre-intervention. Values are mean ± SD. BMI: body mass index, d-ROMs: diacron-reactive oxygen metabolites, BAP: biological antioxidant potential, CK: creatine kinase, LDH: lactate dehydrogenase.

**Table 3 sports-09-00037-t003:** Comparison of pre- and post-intervention oxidative stress markers, CK and LDH in Study 1.

	Intake Group (n = 15)		Control Group (n = 15)	
	Pre	Post	Pre	Post
d-ROMs (U.CARR)	270 ± 35	247 ± 25 **	249 (225–286)	244 (225–292)
BAP (µmol/L)	2013 ± 149	2083 ± 150	2059 ± 142	2143 ± 140
BAP/d-ROMs ratio	7.1 (6.9–8.2)	8.7 (7.3–9.8) *	8.1 ± 1.2	8.5 ± 1.4
CK (U/L)	702 ± 398 ^†^	909 ± 605	330 (195–605)	502 (323–857)
LDH (U/L)	244 (236–269) ^†^	253 (202–269)	222 ± 40	237 ± 54

Values are mean ± SD or median (25–75 percentile). pre-intervention (Pre) vs post-intervention (Post). BAP/d-ROMs ratio and LDH in the Intake group and d-ROMs and CK in the Control group are paired *t*-test, others are Wilcoxon signed-rank test (* *p* < 0.05, ** *p* < 0.01). ^†^ indicates a significant difference as compared to the values of control group (*p* < 0.05). d-ROMs: diacron-reactive oxygen metabolites, BAP: biological antioxidant potential, CK: creatine kinase, LDH: lactate dehydrogenase.

## Data Availability

The data presented in this study are available on request from the corresponding author. The data are not publicly available due to privacy reasons.

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
