# Peer review of "The Intake of Kiwifruits Improve the Potential Antioxidant Capacity in Male Middle- and Long-Distance Runners Routinely Exposed to Oxidative Stress in Japan"

_sports, 2021, doi:10.3390/sports9030037_

Round 1

Reviewer 1 Report

Major Comments:

In this manuscript, the authors investigated the effects of high-antioxidant fruits (kiwifruit) intake on oxidative stress and antioxidant activity in middle- and long-distance runners routinely exposed to oxidative stress. They suggested that the intake of kiwifruits may improve the potential antioxidant capacity. I think this study has strong limits but the authors are aware of this such as described in the final part of discussion. However, I have an important question that have to be clarify by the authors.   

  • Why did the authors attribute the reduced d-ROM values in the study 1 to kiwifruits intake? It is reported that the levels of markers of muscle lipid and protein oxidation are reduced by training while antioxidant enzymes activity are increased (Venditti et al, 2014). So could running be responsible for the adaptive muscle response and the reduced d-ROM values, being a regular aerobic activity?

Minor Comments:

  1. In the Introduction section, lines 32-336, the authors should define the reactive oxygen species (ROS) and when an oxidative stress condition occurs (I suggest reading Di Meo et al., 2019; Venditti et al., 2015). They talk about oxidative stress without defining it (line 36).
  2. In the Introduction section, the references 3-5 describe only the negative influence of Vitamin C on exercise performance. Please also add vitamin E effects on the traininig (I suggest reading Venditti et al., 2014; Venditti et al., 2015).
  3. “Materials and methods, Subjects”: please underlie that the subjects continue to run during the intervention. I think that in the text this concept is not so clear.
  4. Lines 85-87: please add references for d-ROM test, BAP test and BAP/d-ROMs ratio.

Line 180: what is the figure 1? I think that there is a mistake and the figure 2 is the 1

Reviewer 2 Report

see attached review

Reviewer 3 Report

The authors describe the effects of consumption of specific kiwi fruits on some indirect measures of antioxidant stress in trained runners. They report some beneficial effects of kiwi fruit consumption. The authors should consider the following comments.

The validity of the findings of this study depends highly on some analytical determinations. Table 1 shows nutritional content of kiwi fruits but no data are presented for actual kiwi fruit ingested. If kiwi fruit is hypothesized to be beneficial, the specific nutrients to exert these propitious action in support of controlling oxidative stress should be stated and measured in the blood of the runners.

Include quality assurance data for all analytical methods (e.g., accuracy and reproducibility).

Whereas the kiwi fruit provides certain anti-oxidant nutrients, there is no assessment of the nutrients in the usual diet of the runners. Did the runners have adequate micronutrient intake without the kiwi fruit? Thus, one may question whether diet, other than the supplemental kiwi fruits, contributed to any reported beneficial effects. Did the runners ingest any dietary supplements or foods fortified with anti-oxidant nutrients?

Was a control (non-Kiwi supplemented) group involved in this study?

The limitations of no nutritional assessment and determination of changes in circulating anti-oxidant nutrients should be stated.

There is considerable research describing Vit C and E intake relative to anti-oxidant status and muscle injury. This issue should receive more detailed discussion as related to the findings on muscle enzymes associated with muscle damage. 

Round 2

Reviewer 1 Report

I think the authors have done a nice revision of their text and now the manuscript is clearer. This work has limitations like many human studies because it is difficult to control daily intake and quantify their physical activities, but the authors highlighted the criticisms in their text.

I suggest publishing this research article on Sports.

Author Response

Thank you very much for providing comments. We are thankful for the time and energy you expended.

Reviewer 2 Report

see attached review

Reviewer 3 Report

The authors adequately addressed my comments.

My concern with this manuscript is the poor experimental design (failure to assess intakes of micronutrients that impact antioxidant status and actual measurement of objective measures of antioxidant status). The authors can not repeat this experiment thus they need to state this as a limitation of their research.
